# Network Synchronization of MACM Circuits and Its Application to Secure Communications

**DOI:** 10.3390/e25040688

**Published:** 2023-04-19

**Authors:** Rodrigo Méndez-Ramírez, Adrian Arellano-Delgado, Miguel Ángel Murillo-Escobar

**Affiliations:** 1Scientific Research and Advanced Studies Center of Ensenada (CICESE), Ensenada 22860, BC, Mexico; 2National Council of Science and Technology (CONACYT), Ciudad de Mexico 03940, CDMX, Mexico; 3Engineering, Architecture and Design Faculty, Autonomous University of Baja California (UABC), Ensenada 22860, BC, Mexico

**Keywords:** complex network synchronization, star coupled network, MACM, chaotic encryption

## Abstract

In recent years, chaotic synchronization has received a lot of interest in applications in different fields, including in the design of private and secure communication systems. The purpose of this paper was to achieve the synchronization of the Méndez–Arellano–Cruz–Martínez (MACM) 3D chaotic system coupled in star topology. The MACM electronic circuit is used as chaotic nodes in the communication channels to achieve synchronization in the proposed star network; the corresponding electrical hardware in the slave stages receives the coupling signal from the master node. In addition, a novel application to the digital image encryption process is proposed using the coupled-star-network; and the switching parameter technique is finally used to transmit an image as an encrypted message from the master node to the slave coupled nodes. Finally, the cryptosystem is submitted to statistical tests in order to show the effectiveness in multi-user secure image applications.

## 1. Introduction

Synchronization has received a lot of interest in applications in different fields [1,2,3,4], and in recent years, chaotic synchronization has received attention in the implementation of private and secure communication systems [1,5,6,7,8,9,10,11,12,13,14]. Confidential information is encrypted into a transmission using a chaotic signal by direct modulation, masking, or other techniques [7,11]. Thus, one widely studied chaotic system is that of the Chua circuit in synchronization and communication applications [15,16,17,18,19]. In optical communication applications, chaotic synchronization is achieved using a transmitter and a receiver to extract hidden information from the transmitted signal; some studies have been reported in [20,21,22,23,24,25,26,27].

Synchronization in complex dynamic networks has direct applications in different fields, as synchronization is carried out in complex dynamic networks using nodes in different topologies which are connected and coupled, and displaying chaotic behavior in their dynamics once synchronization is achieved, as can be seen in e.g., [9,13,28,29,30,31,32,33,34,35]. Additionally, the synchronization of the two-well Duffing equation is reported using two unidirectionally and bidirectionally coupled piecewise linear maps [36], and the chaotic behavior of the two-well Duffing equation with forcing is computed using the topological entropy—studies of the kneading theory are conducted for symmetric unimodal maps and bimodal maps [37]. Furthermore, the synchronization of complex network spaces has recently been studied in network topologies and explicitly determined in regular ring-lattices [38], whilst networks of discontinuous piecewise linear maps with different slopes [39] are characterized by circulating matrices and the conditional Lyapunov exponents.

Star-coupled networks have several applications in the transmission of information which are implemented using nodes in network coupling, where the master node is the transmitter and it provides a digital message to multiple receivers as slave nodes, e.g., using electronic and optical circuits [31,40], and the experimental proofs to validate the chaos existence using optical circuit communications is proposed in [24].

The objective of this work was to obtain network coupling synchronization using the chaotic MACM circuits in star topology [41]. A single-master circuit with four slave circuits is proposed using the complex systems theory, and the continuous version of the proposed network coupling and the simulation of the electronic circuit simulation is conducted using the Proteus 8 Professional of Labcenter Electronics. The MATLAB tools were used to transmit encrypted confidential digital messages from a single transmitter to multiple receivers, and the digital message process is validated using security analysis such as histograms, correlation analysis, and information entropy.

The organization of the paper is as follows: In Section 2, a brief review on network synchronization theory is provided. In Section 3, a mathematical model of the MACM circuit being used like nodes is described. Section 4 shows the mathematical model of the star coupled network and its synchronization using the MACM circuit, and we show the physical implementation and experimental results for chaos network synchronization using MATLAB and analog circuit simulation as communication channels. In Section 5, we apply the results obtained in the experimental network synchronization to transmit encrypted information from a transmitter to multiple receivers and the security analysis to validate the digital image as a message. Finally, the conclusions are given in Section 6.

## 2. Brief Review on Synchronization of Complex Networks

In this section, a brief review on complex dynamical networks is given, particularly on star coupling topology and its synchronization.

### 2.1. Synchronization of Complex Network

We consider a *complex network* as composed of *N* identical nodes, which are linearly and diffusively coupled through the first state of each node. In this network, each node constitutes an *n-dimensional dynamical system*, described as follows
(1)$${\dot{x}}_{i}=f\left(x_{i}\right)+u_{i},\;\;\;\;\;i=1,2,\ldots ,N,$$
where $x_{i}={\left(x_{i1},\;x_{i2},\;\ldots ,\;x_{in}\right)}^{T}\in R^{n}$ are the *state* variables of the node *i*, $u_{i}=u_{i1}\in R$ is the *input* signal of the node *i*, and is defined by
(2)$$u_{i1}=k\sum_{j=1}^{N}a_{ij}\Gamma x_{j},\;\;\;\;\;i=1,2,\ldots ,N,$$
the constant k>0 represents the *coupling strength* of the complex network, and $\Gamma \in R^{n\times n}$ is a constant 0–1 matrix linking coupled state variables. For simplicity, assume that $\Gamma =diag\left(r_{1},\;r_{2},\ldots ,r_{n}\right)$ is a diagonal matrix with $r_{i}=1$ for a particular *i* and $r_{j}=0$ for j≠i. This means that two coupled nodes are linked through their *i*-th state variables. Whereas, $A=\left(a_{ij}\right)\in R^{N\times N}$ is the *coupling matrix*, which represents the coupling topology of the complex network. If there is a connection between node *i* and node *j*, then $a_{ij}=1$; otherwise, $a_{ij}=0$ fori≠j. The diagonal elements of the coupling matrix A are defined as
(3)$$a_{ii}=-\sum_{j=1,\;j\neq i}^{N}\;a_{ij}=-\sum_{j=1,\;j\neq i}^{N}\;a_{ji},\;\;\;\;\;i=1,2,\ldots ,N.$$
If the degree of node *i* is $d_{i}$, then $a_{ii}=-d_{i},$i=1,2,…,N.

Now, suppose that the complex network is connected without isolated clusters. Then, A is a symmetric irreducible matrix. In this case, it can be shown that zero is an eigenvalue of A with a multiplicity of 1 and all the other eigenvalues of A are strictly negative [29,30].

The synchronization state of nodes in complex systems can be characterized by the nonzero eigenvalues of A. The complex networks (Equation 1) and (Equation 2) are said to (asymptotically) achieve **synchronization** if [30]:(4)$$x_{1}\left(t\right)=x_{2}\left(t\right)=\;\ldots \;=x_{N}\left(t\right),\;\;as\;\;t\to \infty .$$

The diffusive coupling condition (Equation 3) guarantees that the synchronization state is a solution, $s\left(t\right)\in R^{n}$, of an isolated node, that is
(5)$$\dot{s}\left(t\right)=f\left(s(t)\right),$$
wheres(t) can be an *equilibrium point*, a *periodic orbit*, or a *chaotic attractor*. Thus, the stability of the synchronization state
(6)$$x_{1}\left(t\right)=x_{2}\left(t\right)=\;\ldots \;=x_{N}\left(t\right)=s\left(t\right),$$
of complex network (Equation 1) and (Equation 2) is determined by the dynamics of an isolated node—function *f* and solution s(t)—the coupling strength *c*, the inner linking matrix Γ, and the coupling matrix A.

### 2.2. Star Coupled Networks

In this work, we consider complex networks (Equation 1) and (Equation 2) as composed of coupled chaotic nodes in star topology. We assume that all the nodes are connected, without self-loops, and without multiple edges between two nodes. The coupling matrix for star-coupled master–slave networks is given by
(7)$$A=\left[\begin{array}{ccccc} 0 & 0 & 0 & \ldots & 0\\ 1 & -1 & 0 & \ldots & 0\\ 1 & 0 & -1 & \ldots & 0\\ \vdots & \ddots & \ddots & \ddots & \vdots \\ 1 & 0 & 0 & \ldots & -1 \end{array}\right].$$

The eigenvalues of A are (0,−1,−1,…,−1). Therefore, the second largest eigenvalue of A is $\lambda_{2}\;=\;-1$, which is unrelated to the size of the network.

### 2.3. Synchronization Analysis Based on Master Stability Function Approach

We consider a simple complex dynamical network consisting of two coupled chaotic continuous-time nonlinear oscillators (Equation 1) in a master–slave configuration. The well-known master stability function approach [2] is the stability analysis used for studying the synchronous solution in (Equation 1). The generic variational equation governing the behavior around the synchronous solution is
(8)$${\dot{\xi }}_{q}=\left[Df\left(s\right)+\zeta_{q}\Gamma \right]\xi_{q}$$
where Df is the local Jacobian of the vector function f evaluated on a (bounded) trajectory s, in this case of the master system, q=0,1,2,…,N−1 with $\zeta_{q}$ being an eigenvalue of the coupling matrix A, with $\zeta_{0}=0$. We calculated the maximum Floquet or Lyapunov exponents $\lambda_{max}$ for the generic variational Equation (Equation 8) as a function of the coupling strengths *k*, where the synchronous state for $\lambda_{max}>0$ is unstable if $\lambda_{max}<0$ the synchronous state is stable. For the computational calculation of the $\lambda_{max}$, we use the programming software Matlab with initial conditions $s\left(0\right)={\left[1.2,1,1\right]}^{T}$.

## 3. MACM Circuit like Node

In this section, we describe the MACM electronic circuit used as a node to construct the network in the star topology. The objective of this paper was to achieve experimental synchronization in this network using OAs like communication channels [41]. The MACM chaotic system is described as follows
(9)$$\left\{\begin{array}{c} \dot{x}\\ \dot{y}\\ \dot{z} \end{array}\begin{array}{c} =\\ =\\ = \end{array}\begin{array}{c} -ax-byz,\\ -x+cy,\\ d-y^{2}-z. \end{array}\right.$$

MACM’s circuit consists of two no. linear multipliers, it has seven terms and four parameters *a*, *b*, *c*, and *d*, $\in R^{+}$, where *b* and *d* are characterized as the bifurcation parameters *a* = 2, *b* = 2, *c* = 0.5, and *d* = 4.

The electronic implementation of the MACM system is given by the following electronic circuit representation:(10)$$\left\{\begin{array}{ccc} \dot{x} & = & \frac{1}{\mathrm{RC}1}\left(-\frac{R}{R1}x-\frac{R}{10R2}yz\right),\\ \dot{y} & = & \frac{1}{\mathrm{RC}2}\left(-x+\frac{R}{R6}y\right),\\ \dot{z} & = & \frac{1}{\mathrm{RC}3}\left(\frac{R}{R9}d-\frac{R}{10R10}y^{2}-z\right). \end{array}\right.$$

From the system (Equation 10), the electronic components are OAs TL084 U2, U3, analog-multipliers AD633 U1, U4, capacitors C1 = C2 = C3 = 10 nF, and the resistors R1 = 500 kΩ, R2 = 47 kΩ, R3 = R4 = R7 = R8 = R12 = R13 = 10 kΩ, R = R5 = R9 = R11 = 1 MΩ, R6 = 2 MΩ, and R10 = 94 kΩ. The bifurcation parameter *d* was fixed in $V_{d}$ = +3.8 V, the circuit is powered with $+V_{cc}$ = +18 V and $-V_{cc}$ = −18 V. Evaluating the proposed electrical components in the system (Equation 10), the set of parameters of system (Equation 9) are conducted using $a=\frac{R}{R1}$ = 2, $b=\frac{R}{10R2}$ = 2.127, $c=\frac{R}{R6}$ = 0.5, and $d=\frac{R}{R9}$ = 1.

Figure 1 shows the electronic circuit of the MACM system (Equation 10), and its circuit simulation is conducted using the Proteus 8 Professional from Labcenter Electronics [42]. Figure 2 shows temporary trajectories and the phase-planes of the system (Equation 10).

As such, the normalized MACM’s circuit is given by
(11)$$\left\{\begin{array}{c} {\dot{x}}_{1}\\ {\dot{x}}_{2}\\ {\dot{x}}_{3} \end{array}\begin{array}{c} =\\ =\\ = \end{array}\begin{array}{c} -ax_{1}-bx_{2}x_{3},\\ -x_{1}+cx_{2},\\ d-x_{2}^{2}-x_{3}. \end{array}\right.$$

## 4. Star Network Synchronization of MACM’s Circuits

In this section, we describe the complex network to be constructed with *N*-coupled MACM’s circuits (Equation 11), which take the following form (according to Equations (Equation 1) and (Equation 2)): (12)$$\begin{array}{ccc} \left[\begin{array}{c} {\dot{x}}_{i1}\\ {\dot{x}}_{i2}\\ {\dot{x}}_{i3} \end{array}\right] & = & \left[\begin{array}{c} -ax_{i1}-bx_{i2}x_{i3}+u_{i1}\\ -x_{i1}+cx_{i2}\\ d-x_{i2}^{2}-x_{i3} \end{array}\right],i=1,2,\ldots ,N, \end{array}$$(13)$$\begin{array}{ccc} u_{i1} & = & k\sum_{j=1}^{N}a_{ij}\Gamma x_{j}. \end{array}$$

If the control signal in (Equation 13) is $u_{i1}\equiv 0$ for i=1,2,…,N, then we have the original set of *N* uncoupled MACM’s circuits, which evolve according to their own dynamics. We consider, for illustrative purposes, in this work, that N=5, i.e., we have five-coupled MACM’s circuit-like nodes to be synchronized in a star-coupling topology and Γ=diag(1,0,0,0,0) and a coupling constant *k*. In particular, we consider a single master node $N_{1}$ and four slave nodes $N_{2}$, $N_{3},$$N_{4},$ and $N_{5},$ for the physical implementation of this network (Equation 12) and (Equation 13), the topology of which is shown in Figure 3.

Five isolated nodes, such as Equation (Equation 11) are considered to be synchronized in a star network with master node $N_{1}$ and slave nodes $N_{2}$, $N_{3}$, $N_{4}$, and $N_{5}$. The master node $N_{1}$ of the dynamical network is arranged as follows
(14)$$\begin{array}{ccc} \left[\begin{array}{c} {\dot{x}}_{11}\\ {\dot{x}}_{12}\\ {\dot{x}}_{13} \end{array}\right] & = & \left[\begin{array}{c} -ax_{11}-bx_{12}x_{13}+u_{11}\\ -x_{11}+cx_{12}\\ d-x_{12}^{2}-x_{13} \end{array}\right], \end{array}$$
(15)$$\begin{array}{ccc} u_{11} & = & 0, \end{array}$$
the first slave node $N_{2}$ is given by
(16)$$\begin{array}{ccc} \left[\begin{array}{c} {\dot{x}}_{21}\\ {\dot{x}}_{22}\\ {\dot{x}}_{23} \end{array}\right] & = & \left[\begin{array}{c} -ax_{21}-bx_{22}x_{23}+u_{21}\\ -x_{21}+cx_{22}\\ d-x_{22}^{2}-x_{23} \end{array}\right], \end{array}$$
(17)$$\begin{array}{ccc} u_{21} & = & k(x_{11}-x_{21}), \end{array}$$
the second slave node $N_{3}$ by means of
(18)$$\begin{array}{ccc} \left[\begin{array}{c} {\dot{x}}_{31}\\ {\dot{x}}_{32}\\ {\dot{x}}_{33} \end{array}\right] & = & \left[\begin{array}{c} -ax_{31}-bx_{32}x_{33}+u_{31}\\ -x_{31}+cx_{32}\\ d-x_{32}^{2}-x_{33} \end{array}\right], \end{array}$$
(19)$$\begin{array}{ccc} u_{31} & = & k(x_{11}-x_{31}), \end{array}$$
the third slave node $N_{4}$ as follows
(20)$$\begin{array}{ccc} \left[\begin{array}{c} {\dot{x}}_{41}\\ {\dot{x}}_{42}\\ {\dot{x}}_{43} \end{array}\right] & = & \left[\begin{array}{c} -ax_{41}-bx_{42}x_{43}+u_{41}\\ -x_{41}+cx_{42}\\ d-x_{42}^{2}-x_{43} \end{array}\right], \end{array}$$
(21)$$\begin{array}{ccc} u_{41} & = & k(x_{11}-x_{41}), \end{array}$$
and the fourth slave node $N_{5}$ given by
(22)$$\begin{array}{ccc} \left[\begin{array}{c} {\dot{x}}_{51}\\ {\dot{x}}_{52}\\ {\dot{x}}_{53} \end{array}\right] & = & \left[\begin{array}{c} -ax_{51}-bx_{52}x_{53}+u_{51}\\ -x_{51}+cx_{52}\\ d-x_{52}^{2}-x_{53} \end{array}\right], \end{array}$$
(23)$$\begin{array}{ccc} u_{51} & = & k(x_{11}-x_{51}). \end{array}$$

Now, using Equations (Equation 14) and (Equation 23) as chaotic nodes, we have constructed the star network with master node $N_{1}$ to be synchronized according to Figure 3. The corresponding coupling matrix is given by
(24)$$A=\left[\begin{array}{ccccc} 0 & 0 & 0 & 0 & 0\\ 1 & -1 & 0 & 0 & 0\\ 1 & 0 & -1 & 0 & 0\\ 1 & 0 & 0 & -1 & 0\\ 1 & 0 & 0 & 0 & -1 \end{array}\right].$$
with eigenvalues $\lambda_{1}=0$, $\lambda_{2}=\lambda_{3}=\lambda_{4}=$ $\lambda_{5}=-1.$

Figure 4 shows $\lambda_{max}$ applying the coupling matrix (Equation 24) for a range 0≤k≤35 using the MACM system (Equation 9) as a node, where the sufficient coupling strength *k* to achieve network synchronization is k>7; therefore, this analysis is used to consider the coupling strengths used in the numerical simulations and in the corresponding electronic implementation.

### 4.1. Synchronization Analysis Based on Master Stability Function Approach and Its Simulation

The chaotic synchronization in a star network topology is implemented in MATLAB for numerical results. We use the ODE45 function to integrate the system of differential equations of first order using the explicit formula Runge–Kutta (4,5) [43]. The control parameters are the same for each MACM system (Equation 9), i.e., a=2, b=2, c=0.5, and d=4, but with different initial conditions for each system presented in Table 1, and the time-series and phase-plane graphics of master MACM system 1 are shown in Figure 5. Thus, all the MACM systems present chaotic behavior.

We define the error synchronization as $e2=x_{11}-x_{21},\;e3=x_{11}-x_{31},\;e4=x_{11}-x_{41},\;e5=x_{11}-x_{51}$. In Figure 6 and Figure 7, the time series of the errors and the phase graphics between the master and the four slaves are presented without coupling, i.e., the coupling constant is defined as zero. The results show that the network is not synchronized. Considering a coupling constant k=10, the four slave MACM systems in the star network synchronize with the master MACM system after the transient time, as presented in Figure 8 with the errors over time, and Figure 9 shows the synchronization graphic by plotting the corresponding phases between the MACM master and the four MACM slaves systems and producing a line with 45 degrees after the transient time. Based on the simulation results, the synchronization is achieved in the three chaotic states after 40 time units.

### 4.2. Star Network Electronic Circuit Synchronization

In this section, we describe the simulation of the electronic circuit implementation of the coupled-star-network and its synchronization to conduct the set-up among the master node ($N_{1}$) and four slave nodes ($N_{2}$, $N_{3}$, $N_{4}$, and $N_{5}$). Based on the arrangement of five coupled MACM circuits, as shownvi in Equations (Equation 14)–(Equation 23) with the coupling signal $u_{11}\equiv 0$ in (), the entire electronic circuit implementation is conducted by the means of analog circuits and passive electrical components which are shown in Figure 10. Table 2 shows the hardware used in the electronic implementation of the coupled-star-network to achieve network synchronization in each channel, the same power supply of Figure 1 was used in this electronic implementation using $V_{d}$ = +3.8 V, $+V_{cc}$ = +18 V and $-V_{cc}$ = −18 V, and the set of 15 capacitors and 97 resistors electronic components, 10 analog multipliers AD633, and 9 OAs TL084 as ICs.

In order to obtain different initial conditions in the electronic simulation of Figure 10, we propose the initial condition settings for each node using electric components with slightly different values; for the slave nodes $N_{2}$, $N_{3}$, $N_{4}$, and $N_{5}$, we used the resistors R24, R50, R29, and R55, respectively.

The chaotic dynamics of the states $x_{11}\left(t\right)$ and $x_{12}\left(t\right)$ and the chaotic attractor ($x_{11}$ versus $x_{12}$) of the electrical simulation corresponding to the master MACM circuit, as shown in Figure 11.

As the state $x_{11}\left(t\right)$ of master $N_{1}$ versus state $x_{21}\left(t\right)$ of slave $N_{2}$ is shown in Figure 12, we can see the chaotic nodes $N_{1}$ and $N_{2}$ without coupling. The synchronization of the MACM circuits of the Equations (Equation 14) and (Equation 23) is achieved using the proposed network circuit depicted in Figure 12, the errors are shown in Figure 13, and the phase-planes are shown in Figure 14. For the other three slave nodes $N_{3}$, $N_{4}$, and $N_{5}$, we used the same process given for the synchronization of $N_{1}$ versus $N_{2}$.

## 5. Application to Image Encryption

The encryption of sensitive data in networks provides privacy to users. Particularly, digital images are transmitted over insecure channels throughout the Internet. The application of chaos synchronization to secure communications was proposed by Pecora and Carrol in 1990 [1].

In contrast with chaos-based cryptography which uses permutation and diffusion to encrypt image data to one receptor [44,45], we present the application of image encryption in a star network to securely transmit a digital image from the master MACM system to four slave MACM systems (multiple receptors) using chaotic synchronization and switching parameter technique [40,46]. In Figure 15, the schematic of the proposed image encryption process is presented. The process to transmit the digital image is described in the next steps:1.**Binary string**. The 8-bit gray-scale digital image with M(row)×N(columns) pixels are placed row-by-row in a binary string with M×N×8 bits.2.**Synchronization of star network**. We used a coupling constant of k=10 between the master and slave MACM systems; different initial conditions are used for each MACM system (see Table 1); the control parameters are the same in all MACM systems, i.e., a=2,b=2,c=0.5, and d=4. After 50 time units (transient time), the star network is synchronized as shown in Figure 16.3.**Extended plain binary data**. Since synchronization is achieved after a transient time and to avoid data loss in the receptors, the plain binary string is mounted over 400 time units for each bit producing an extended plain binary data of M×N×8×400. As an example, Figure 16a–d show the first two bytes of the plain image transmitted, which are defined as 1010010010100011 with a length of 6400 time units (dashed line).4.**Switching parameter *d* of master MACM**. The parameter *d* of the master node is switched between d=4 and d=4.05, for 0 and 1 in the extended plain binary data, respectively. During this time, the absolute synchronization error is determined in e2, e3, e4, and e5, which are shown in Figure 16a–d with a blue line. Since initial conditions are considerably different at the start communication, the error is bigger in the first time units.5.**Processing the error**. The recovered binary string in the receptor is calculated with the sum of the last 100 data in each error signal considering windows of 400 data; if the sum is greater than 0.7, a bit of 1 is defined for such window or bit of 0 in other case. Figure 16e–h presents the first recovered binary string in each slave MACM system (receptor).6.**Image construction**. The digital image is constructed using the recovered binary string and the inverse process of step 1; the string is separated into 8-bit segments and assigned to rows and columns to form the corresponding digital image. Figure 16i–l present the difference between the plain image and recovered image at the bit level (first 8000 bits) for slaves 2–5, respectively.

The proposed image encryption process is implemented at the software level in MATLAB (R2015a) in one laptop with a Intel Core 2.9 GHz processor, 8 GB of RAM, and operative system Windows 10 of 64 bits. The results of image encryption and decryption with the coupling constant k=10 are presented in Figure 17. Figure 17a shows the plain image of Lena with 150×150 pixels to be transmitted by the master MACM; Figure 17b shows the cryptogram, which is constructed with the chaotic signal $x_{11}$ as a noise image; Figure 17c–f present the decrypted image in slaves 1–4, respectively. When the network is not synchronized, the images cannot be recovered in the slaves.

**Figure 16 entropy-25-00688-f016:**
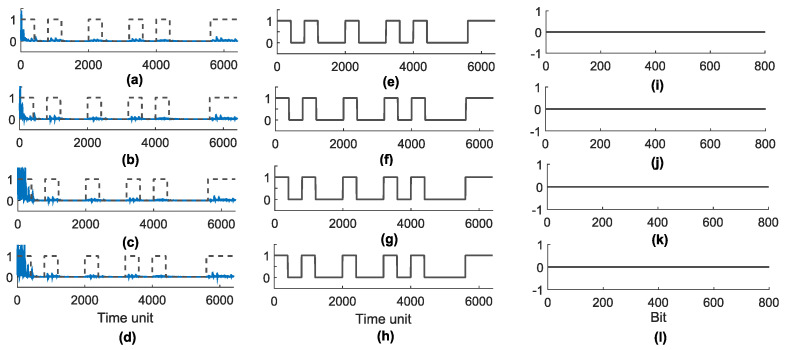
First bytes transmitted and recovered data: (**a**) absolute of $e_{2}$ (blue line) and plain binary data (dashed line); (**b**) absolute of $e_{3}$ (blue line) and plain binary data (dashed line); (**c**) absolute of $e_{4}$ (blue line) and plain binary data (dashed line); (**d**) absolute of $e_{5}$ (blue line) and plain binary data (dashed line); (**e**) recovered data in slave 2; (**f**) recovered data in slave 3; (**g**) recovered data in slave 4; (**h**) recovered data in slave 5; (**i**) error between plain image end recovered image in slave 2; (**j**) error between plain image end recovered image in slave 3; (**k**) error between plain image end recovered image in slave 4; and (**l**) error between plain image end recovered image in slave 5.

On the other hand, the results of image encryption and decryption with coupling constant k=5 is presented in Figure 18. Figure 18a shows the plain image of Lena with 150×150 pixels to be transmitted by the master MACM; Figure 18b shows the cryptogram, which is constructed with the chaotic signal $x_{11}$ as a noise image; and Figure 18c–f present the decrypted image in slaves 1–4, respectively. Since slaves nodes do not synchronize with the master node, the Lena image cannot be recovered correctly.

### 5.1. Security Analysis

In this subsection, we present the security analysis such as histograms, correlation, and entropy tests to show the effectiveness of the propose encryption mechanism.

#### 5.1.1. Histograms

In the histogram attack, the cryptanalyst attempts to find a statistical relation with plain text. It must be uniform to resist such an attack. In Figure 19 and Figure 20, the histograms of Figure 17 and Figure 18 are presented, respectively. The plain Lena image in all cases has its particular histogram curves, whereas the encrypted image and the incorrect recovered images have a uniform data distribution. Thus, the proposed schema can resist a histogram attack.

#### 5.1.2. Statistics of Histogram

The statistics of histograms are evaluated using the variance and standard deviation as metrics of data dispersion, i.e., such metrics provide information about variations in a dataset. First, the variance measures the average difference with respect the mean called $\bar{m}$. The more uniform the histogram is, the lower the variance is. The variance is calculated with the following expression
(25)$$\alpha =\frac{1}{256}\sum_{i=1}^{256}{\left(m_{i}-\bar{m}\right)}^{2},$$
where
(26)$$\bar{m}=\frac{M\times N}{256},$$
and where *m* is the frequency in the histogram, α is the variance, *M* is the rows of image, *N* is the columns of the image, and $\bar{m}$ is the mean of the histogram. On the other hand, the standard deviation provides information about the fluctuations versus the mean and it is calculated as follows
(27)$$\beta =\sqrt{\alpha },$$
with β as the standard deviation.

Table 3 presents the variance and the standard deviation for Figure 17 (correct decryption in slaves with k=10) and Figure 18 (incorrect decryption in slaves with k=5) for the plain and encrypted image of Lena with 150 pixels. The plain image presents high variance and standard deviation since data in the histogram are not uniform with fluctuations of 60 around the mean. The uniformity of histograms in encrypted image reduces considerably both metrics achieving a variance of 101.04 with fluctuations of just 10 around the mean.

#### 5.1.3. Structural Similarity Index

The structural similarity index (*SSIM*) is used to determine the similarity structurally between the plain image and the encrypted image. *SSIM* uses the mean, standard deviation, and the cross-correlation of two images *P* and *E*. *SSIM* is evaluated as follows
(28)$$SSIM=\frac{\left(2\bar{P}\bar{E}+T_{1}\right)\left(2\sigma_{PE}+T_{2}\right)}{\left({\bar{P}}^{2}+{\bar{E}}^{2}+T_{1}\right)\left(\beta_{P}^{2}+\beta_{E}^{2}+T_{2}\right)},$$
where
(29)$$\sigma_{PE}=\frac{1}{M\times N}\;\sum_{i=1}^{M}\sum_{j=1}^{N}\left[P\left(i,j\right)-\bar{P}\right]\;\left[E\left(i,j\right)-\bar{E}\right],$$
and $\bar{P}$ is the mean of the plain image, $\bar{E}$ is the mean of encrypted image, $\beta_{P}$ is the standard deviation of plain image, $\beta_{E}$ is the standard deviation of encrypted image, $\sigma_{PE}$ is the cross-correlation of plain and encrypted image, and SSIM≤1. $T_{1}={\left(W_{1}\;L\right)}^{2}$ and $T_{2}={\left(W_{2}\;L\right)}^{2}$ are used for stability, where L=255 (for gray-scale) is the dynamic range of the pixel values with $W_{1}=0.01$ and $W_{2}=0.03$.

Table 4 presents the *SSIM* for Figure 17 (correct decryption in slaves with k=10) and Figure 18 (incorrect decryption in slaves with k=5). If both tested images are identical, the *SSIM=1*. A value of *SSIM* close to zero means both tested images are structurally different, as expected between plain image and correctly decrypted images in slaves (k=10). Nevertheless, the *SSIM* is close to 1 if k=5 since the system does not synchronize and the images cannot be recovered in slaves. Thus, the *SSIM* is close to 1 in such cases.

#### 5.1.4. Correlation Analysis

Plain images present high correlation between the neighbors pixels that must be eliminated in encrypted images to provide security and reduce the risk of statistical attacks. It can be visually observed by using a graphical image correlation and calculating the Pearson correlation coefficient [47].

First, the value between two pixels in any direction of the plain image is similar and plotting the graphic correlation produces several points over the 45 degree line. On the other hand, the encrypted image plot datas in all the space of the graphical correlation, which means that two pixels proximate to one another are different in amplitude. In Figure 21 and Figure 22, the graphic correlation of Figure 17 and Figure 18 are presented, respectively. The plain image and the retrieved images in the five slaves graphically show high correlation between the proximate pixels, whereas the encrypted image presents different pixel values between the neighbors. In Table 5, the Pearson correlation coefficient is presented.

#### 5.1.5. Information Entropy

The information entropy is a metric to numerically determine the level of randomness in images. Since plain images are based on 8-bit data, the maximum entropy is eight [48]. In Table 6, the entropy results of Figure 21 and Figure 22 are presented, respectively. The entropy value close to 8 in an encrypted image means a highly unpredictable message, whereas lower entropy is expected in plain images.

#### 5.1.6. Decryption Error Test

In several applications in chaos-based image encryption such as in telemedicine or biometric systems, the decrypted image must be identical to an encrypted image. Thus, the error between both images must be determined quantitatively, where the plain image and the decrypted image are compared pixel-by-pixel. Based on [49], the decryption error is defined as follows
(30)$$E\left(\%\right)=\frac{100}{M\times N}\sum_{i=1}^{M}\sum_{j=1}^{N}Q\left(i,j\right)$$
and
(31)$$Q\left(i,j\right)=\left\{\begin{array}{cc} 0 & ifP(i,j)=D(i,j)\\ 1 & ifP(i,j)\neq D(i,j) \end{array}\right.$$
and *P* is the original plain image, the *D* is the decrypted image, and E is the error calculated in percentage. In Table 7, the errors in percentage between the plain image and the decrypted images in the slaves are presented. When the coupling constant is k=10, just the 0.0044% of the pixels are lost, i.e., 8 bits of 180000. On the other hand, when the slaves do not synchronize with the master system (k=5), the error the in decrypted image is close to 100%.

## 6. Conclusions

In this study, the simulation of the network synchronization among one-master and four-slave chaotic MACM-systems was conducted by means of complex systems theory. The electronic circuit of the MACM-system was carried-out using the Proteus 8 Labcenter Electronics as an electrical circuit simulator to achieve the coupled-star-network synchronization, the set of one-master and four-slave nodes were implemented using simple integrated circuits, such as operational amplifiers, analog multipliers, and passive components. In addition, the application of the secure communication was conducted in the MATLAB simulation to transmit a digital image message encrypted from a chaotic transmitter to four chaotic receivers, the coupled-star-network synchronization showed good performance in the security analysis results, such as an uniform histogram, high correlation between neighbors, and low performance in the tests of information entropy and decryption error. Finally, the simulation results of the electronic circuits implementation and secure communication showed good performance in the synchronization of the chaotic coupled-star network of the MACM system for encrypting, transmitting, and recovering the secret messages. In future work, we will conduct the digital implementation of the coupled-star-network synchronization of the MACM chaotic system using embedded systems.

## Figures and Tables

**Figure 1 entropy-25-00688-f001:**
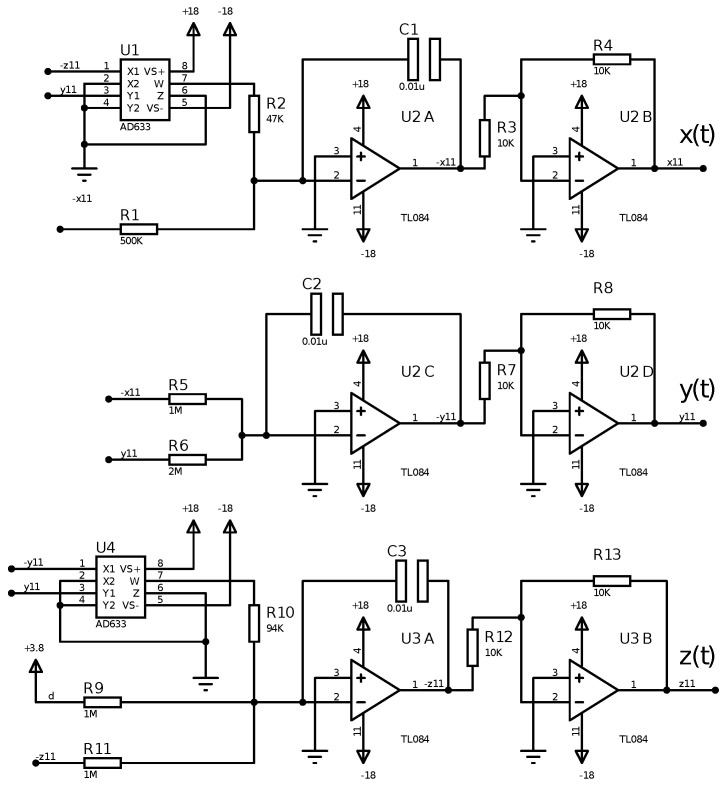
Electronic circuit of the MACM’s system (Equation 10).

**Figure 2 entropy-25-00688-f002:**
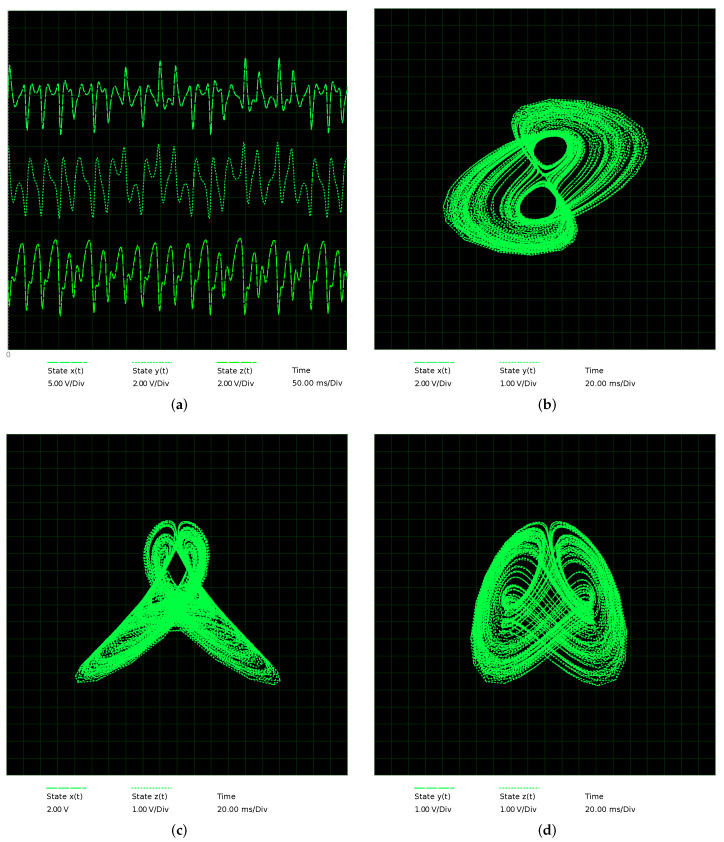
Electronic circuit simulation of the chaotic 3D MACM system (1): (**a**) time evolution of states x(t), y(t), and z(t); (**b**) phase plane x(t) versus y(t); (**c**) phase plane x(t) versus z(t); and (**d**) phase plane y(t) versus z(t).

**Figure 3 entropy-25-00688-f003:**
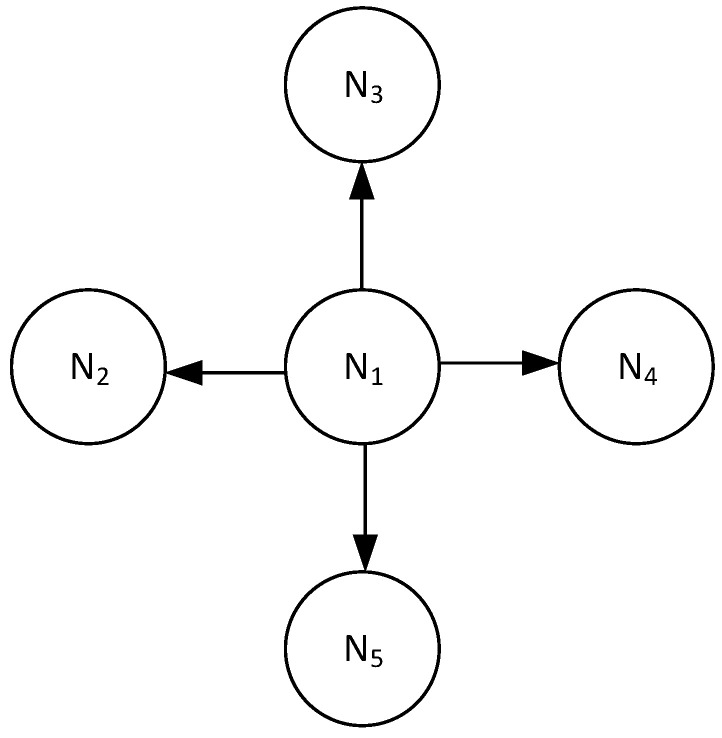
Star network with master node $N_{1}$ and four slave nodes.

**Figure 4 entropy-25-00688-f004:**
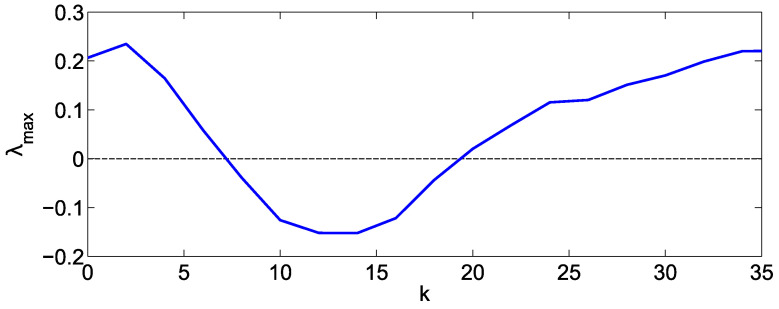
Maximum Lyapunov exponent $\lambda_{max}$ applying coupling matrix A for 0≤k≤35.

**Figure 5 entropy-25-00688-f005:**
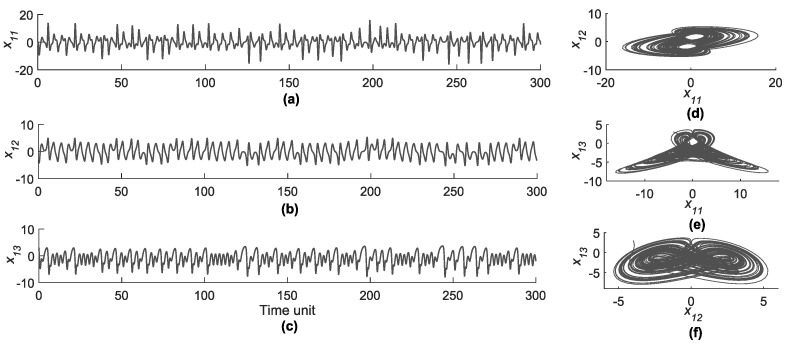
Master MACM system in chaotic regime: (**a**) *x* versus time; (**b**) *y* versus time; (**c**) *z* versus time; (**d**) *x* versus *y*; (**e**) *x* versus *z*; and (**f**) *y* versus *z*.

**Figure 6 entropy-25-00688-f006:**
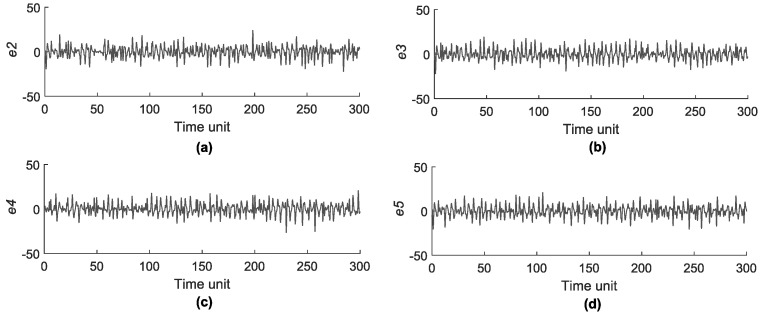
Time series of the errors for each MACM system in the star network without coupling: (**a**) e2; (**b**) e3; (**c**) e4; and (**d**) e5.

**Figure 7 entropy-25-00688-f007:**
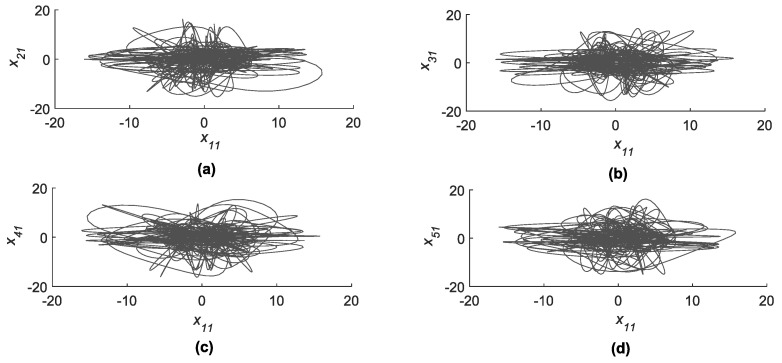
Phase graphics between the master MACM and slaves in the star network without coupling: (**a**) $x_{11}$ versus $x_{21}$; (**b**) $x_{11}$ versus $x_{31}$; (**c**) $x_{11}$ versus $x_{41}$; and (**d**) $x_{11}$ versus $x_{51}$.

**Figure 8 entropy-25-00688-f008:**
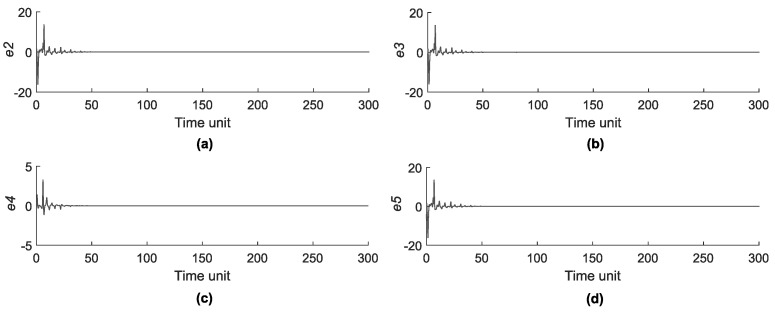
Time series of the errors for each MACM system in the star network with coupling constant k=10: (**a**) e2; (**b**) e3; (**c**) e4; and (**d**) e5.

**Figure 9 entropy-25-00688-f009:**
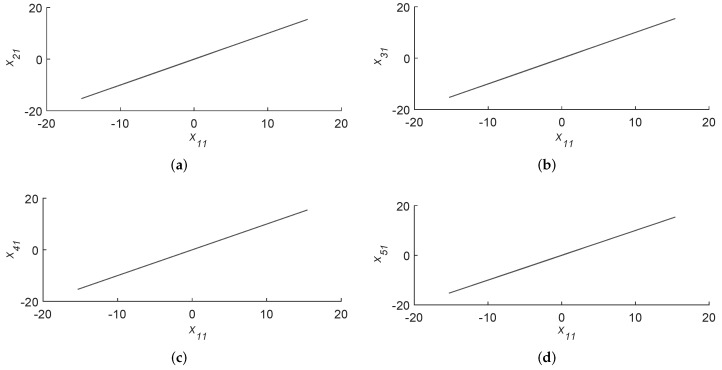
Phase graphics between the master MACM and slaves in the star network with coupling constant k=10: (**a**) $x_{11}$ versus $x_{21}$; (**b**) $x_{11}$ versus $x_{31}$; (**c**) $x_{11}$ versus $x_{41}$; and (**d**) $x_{11}$ versus $x_{51}$.

**Figure 10 entropy-25-00688-f010:**
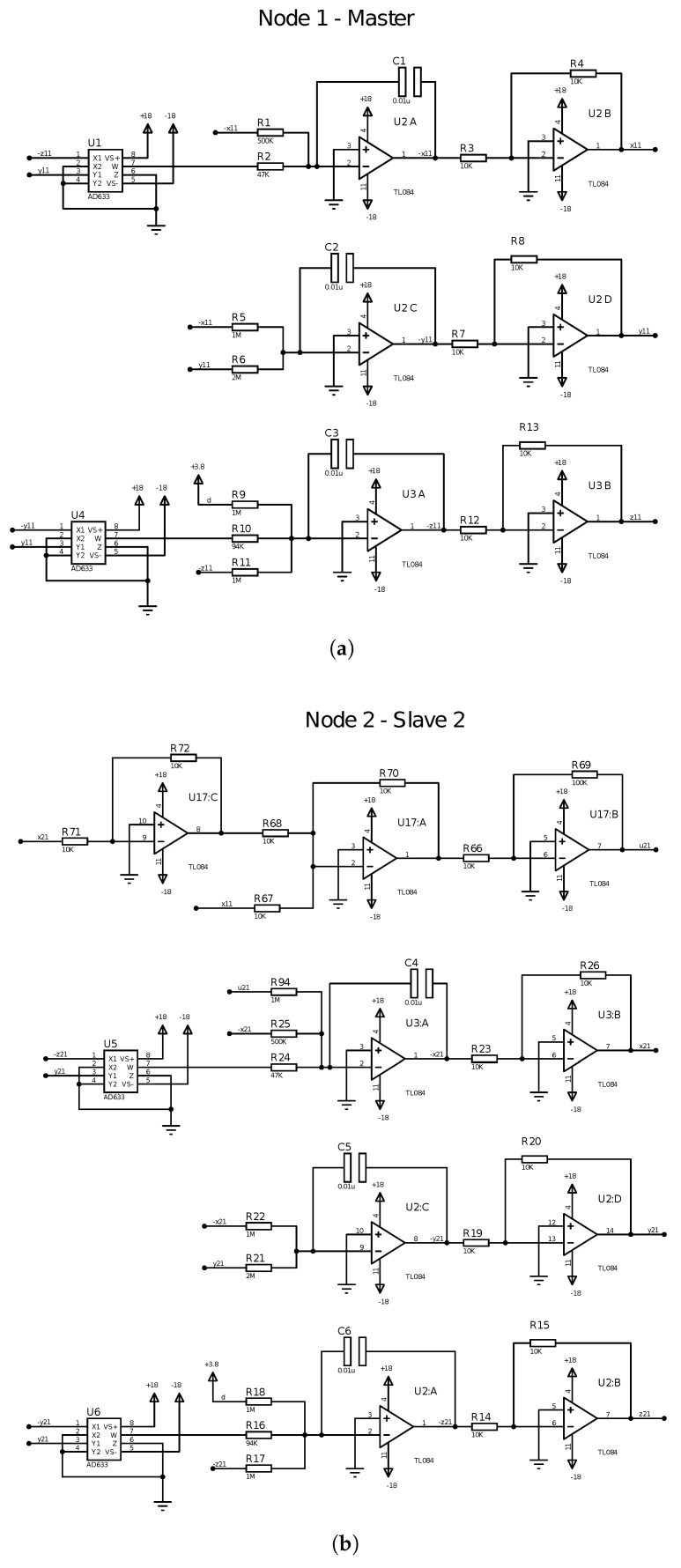
Experimental set-up for network synchronization of five coupled MACM electronic circuits in star topology: (**a**) master node $N_{1}$; (**b**) slave node $N_{2}$; (**c**) slave node $N_{3}$; (**d**) slave node $N_{4}$; and (**e**) slave node $N_{5}$.

**Figure 11 entropy-25-00688-f011:**
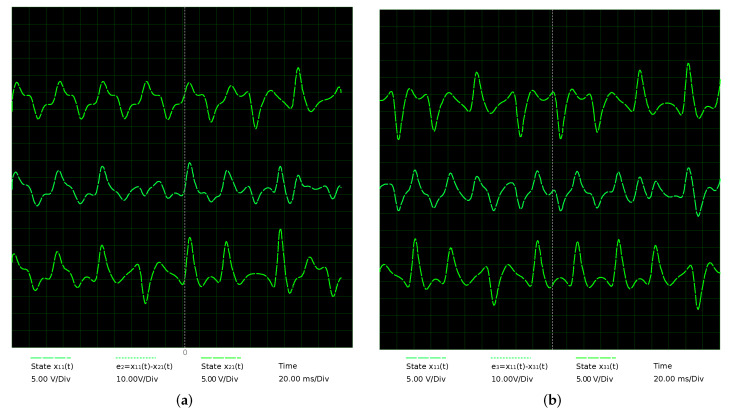
Chaotic trajectories of the system of the Equations (Equation 14) and (Equation 23): (**a**) state $x_{11}\left(t\right)$, error $e2\left(t\right)=x_{11}\left(t\right)-x_{21}\left(t\right)$, and state $x_{21}\left(t\right)$; (**b**) state $x_{11}\left(t\right)$, error $e3\left(t\right)=x_{11}\left(t\right)-x_{31}\left(t\right)$, and state $x_{31}\left(t\right)$; (**c**) state $x_{11}\left(t\right)$, error $e4\left(t\right)=x_{11}\left(t\right)-x_{41}\left(t\right)$, and state $x_{41}\left(t\right)$; and (**d**) state $x_{11}\left(t\right)$, error $e5\left(t\right)=x_{11}\left(t\right)-x_{51}\left(t\right)$, and state $x_{51}\left(t\right)$.

**Figure 12 entropy-25-00688-f012:**
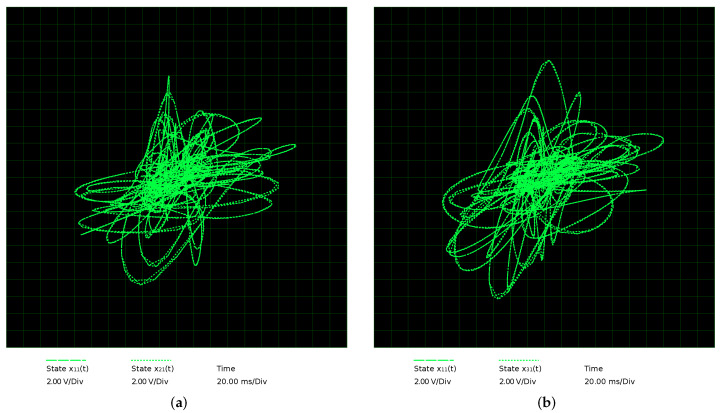
Plane phase of the system of Equations (Equation 14) and (Equation 23): (**a**) $x_{21}\left(t\right)$ versus $x_{11}\left(t\right)$; (**b**) $x_{11}\left(t\right)$ versus $x_{31}\left(t\right)$; (**c**) $x_{11}\left(t\right)$ versus $x_{41}\left(t\right)$; and (**d**) $x_{11}\left(t\right)$ versus $x_{51}\left(t\right)$.

**Figure 13 entropy-25-00688-f013:**
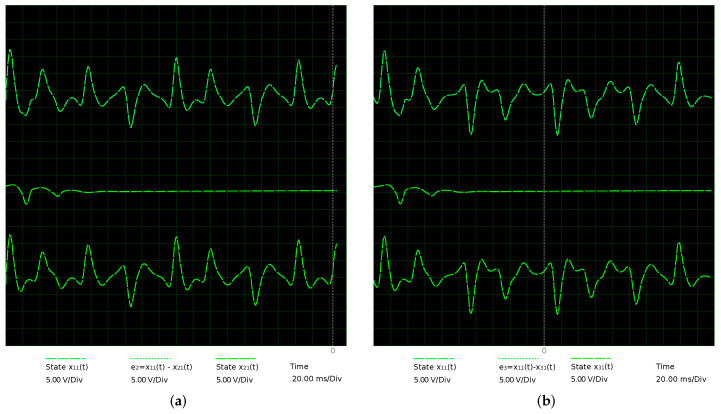
Chaotic trajectories of Equations (Equation 14) and (Equation 23): (**a**) state $x_{11}\left(t\right)$, error $e2\left(t\right)=x_{11}\left(t\right)-x_{21}\left(t\right)$, and state $x_{21}\left(t\right)$; (**b**) state $x_{11}\left(t\right)$, error $e3\left(t\right)=x_{11}\left(t\right)-x_{31}\left(t\right)$, and state $x_{31}\left(t\right)$; (**c**) state $x_{11}\left(t\right)$, error $e4\left(t\right)=x_{11}\left(t\right)-x_{41}\left(t\right)$, and state $x_{41}\left(t\right)$; and (**d**) state $x_{11}\left(t\right)$, error $e5\left(t\right)=x_{11}\left(t\right)-x_{51}\left(t\right)$, and state $x_{51}\left(t\right)$.

**Figure 14 entropy-25-00688-f014:**
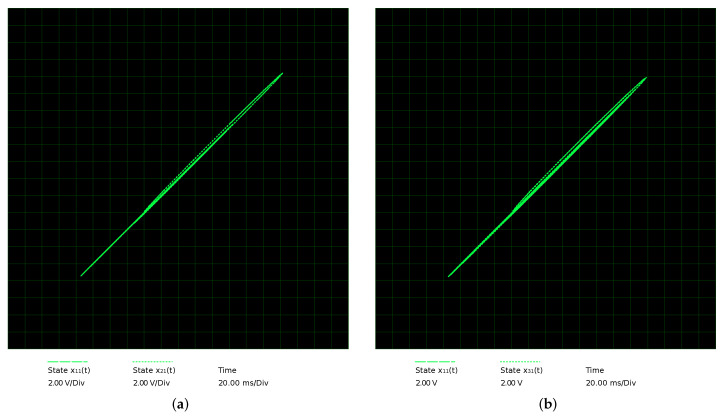
Plane phase of Equations (Equation 14) and (Equation 23): (**a**) $x_{21}\left(t\right)$ versus $x_{11}\left(t\right)$; (**b**) $x_{11}\left(t\right)$ versus $x_{31}\left(t\right)$; (**c**) $x_{11}\left(t\right)$ versus $x_{41}\left(t\right)$; and (**d**) $x_{11}\left(t\right)$ versus $x_{51}\left(t\right)$.

**Figure 15 entropy-25-00688-f015:**
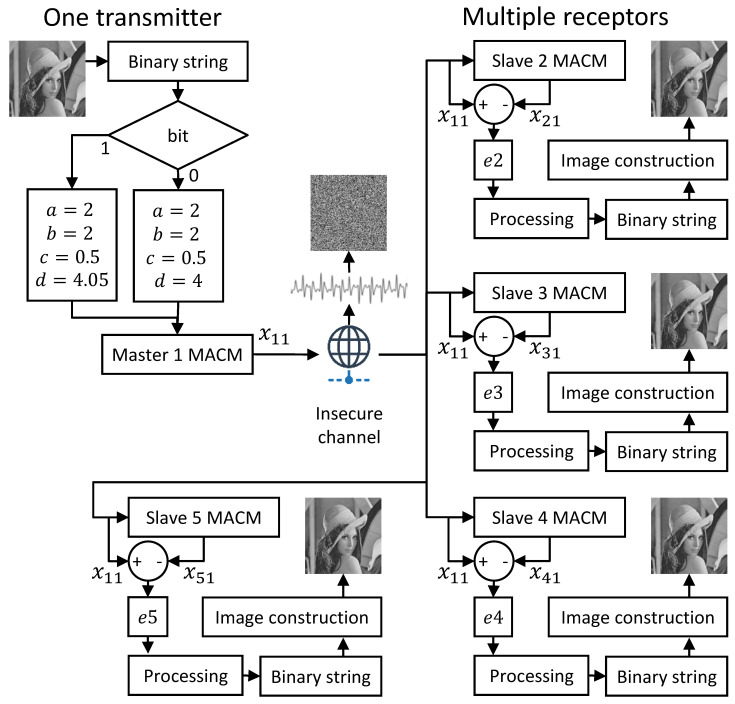
Schematic of the proposed image encryption process.

**Figure 17 entropy-25-00688-f017:**
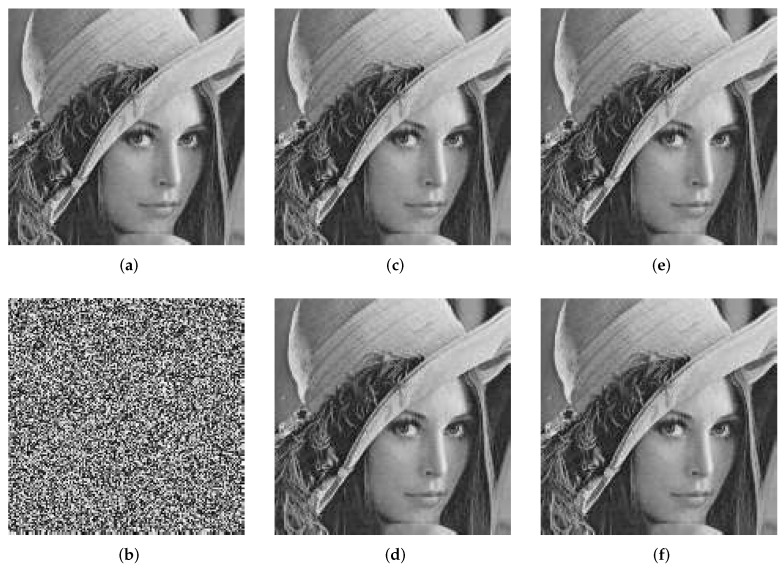
Experimental results of image encryption with the coupling constant k=10: (**a**) plain Lena image; (**b**) cryptogram; (**c**) decrypted image in slave 2 MACM; (**d**) decrypted image in slave 3 MACM; (**e**) decrypted image in slave 4 MACM; and (**f**) decrypted image in slave 5 MACM.

**Figure 18 entropy-25-00688-f018:**
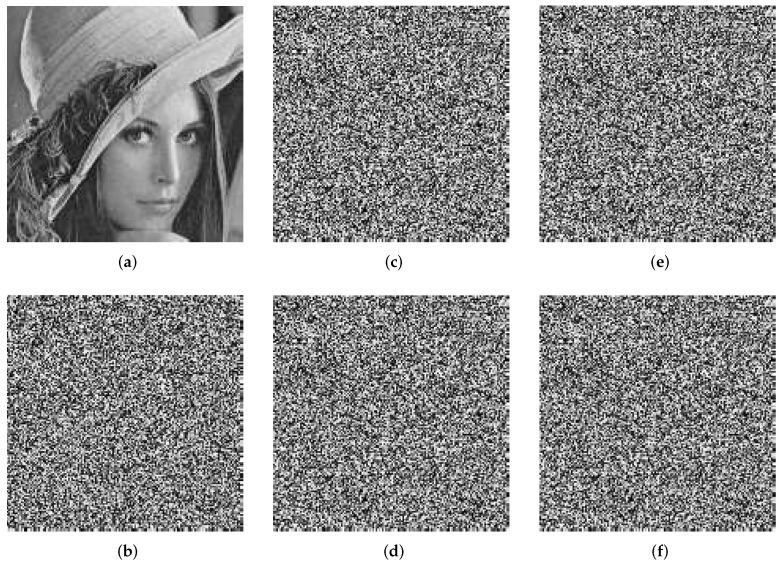
Experimental results of image encryption with coupling constant k=5: (**a**) plain Lena image; (**b**) cryptogram; (**c**) decrypted image in slave 2 MACM; (**d**) decrypted image in slave 3 MACM; (**e**) decrypted image in slave 4 MACM; and (**f**) decrypted image in slave 5 MACM.

**Figure 19 entropy-25-00688-f019:**
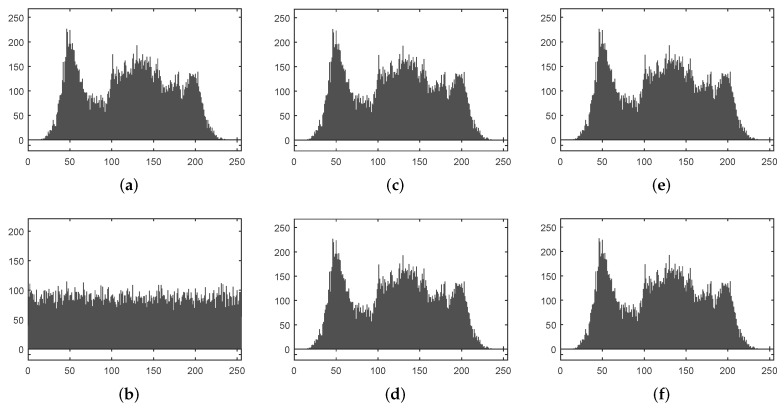
Histograms of images with the coupling constant k=10: (**a**) plain Lena histogram; (**b**) histogram of cryptogram; (**c**) histogram in slave 2 MACM; (**d**) histogram in slave 3 MACM; (**e**) histogram in slave 4 MACM; and (**f**) histogram in slave 5 MACM.

**Figure 20 entropy-25-00688-f020:**
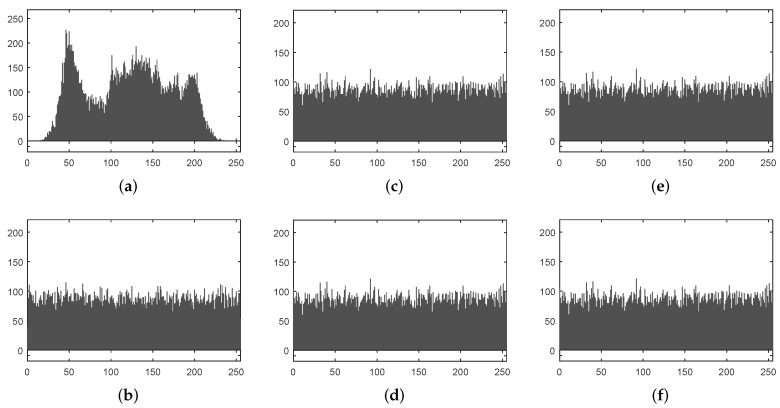
Histograms of images with coupling constant k=5: (**a**) plain Lena histogram; (**b**) histogram of cryptogram; (**c**) histogram in slave 2 MACM; (**d**) histogram in slave 3 MACM; (**e**) histogram in slave 4 MACM; and (**f**) histogram in slave 5 MACM.

**Figure 21 entropy-25-00688-f021:**
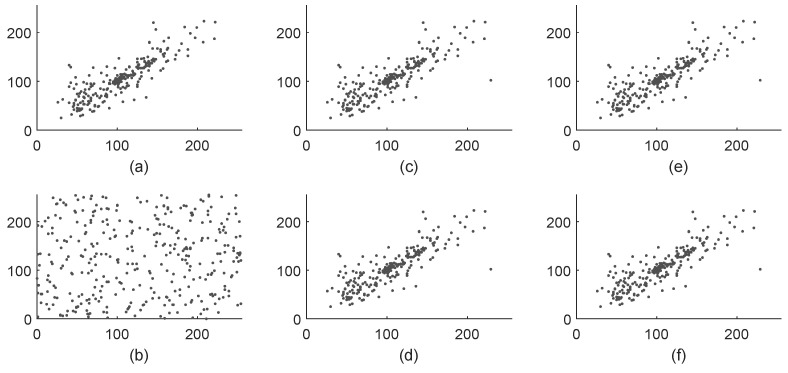
Graphic correlation of images with the coupling constant k=10: (**a**) plain Lena histogram; (**b**) cryptogram; (**c**) correlation in slave 2 MACM; (**d**) correlation in slave 3 MACM; (**e**) correlation in slave 4 MACM; (**f**) correlation in slave 5 MACM.

**Figure 22 entropy-25-00688-f022:**
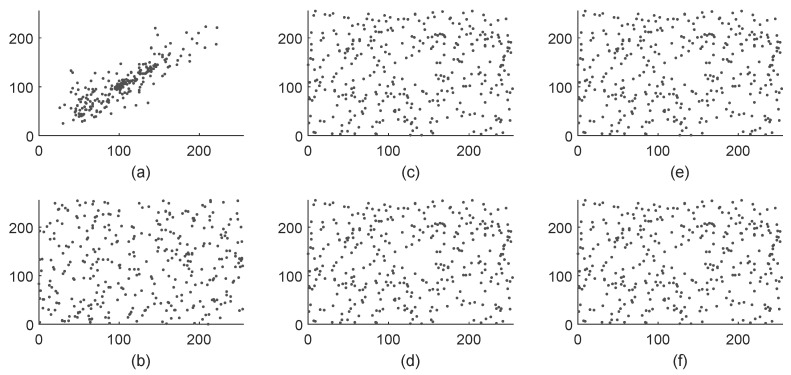
Graphic correlation of images with coupling constant k=5: (**a**) plain Lena histogram; (**b**) correlation of cryptogram; (**c**) correlation in slave 2 MACM; (**d**) correlation in slave 3 MACM; (**e**) correlation in slave 4 MACM; (**f**) correlation in slave 5 MACM.

**Table 1 entropy-25-00688-t001:** Initial conditions of five master MACMs for numerical results in MATLAB.

Initial	Master 1	Slave 2	Slave 3	Slave 4	Slave 5
Condition	MACM	MACM	MACM	MACM	MACM
$x_{i1}\left(0\right)$	−4.0	2.0	2.5	−4.5	2.2
$x_{i2}\left(0\right)$	−4.0	2.0	2.5	−4.5	2.2
$x_{i3}\left(0\right)$	−3.0	4.0	4.5	−3.5	4.2

**Table 2 entropy-25-00688-t002:** Hardware description of the coupled-star-network to achieve network synchronization, as depicted in Figure 10.

Component or IC	Value or Description
C1, C2, C3, C4, C5, C6, C7, C8, C9, C10, C11, C12, C13, C14, C15	10 nF
R1, R25, R38, R51, R64	500 kΩ
R2, R37, R63	47 kΩ
R3, R4, R7, R8, R12, R13, R14, R15, R19, R20, R23, R26, R27, R28, R32, R33, R36, R39, R40, R41, R45, R46, R49, R52, R53, R54, R58, R59, R62, R66, R66, R67, R68, R69, R70, R71, R72, R73, R74, R75, R76, R77, R78, R79, R80, R81, R82, R83, R84, R85, R86, R87 R88, R89, R90, R91, R92, R93	10 kΩ
R5, R9, R11, R17, R18, R22, R31, R30, R35, R43, R44, R48, R56, R57, R61, R94, R95, R96, R97	1 MΩ
R6, R21, R34, R47, R60	2 MΩ
R10, R16, R42	94 kΩ
R24	47.5 kΩ
R50	48 kΩ
R29	94.5 kΩ
R55	95 kΩ
U1, U4, U5, U6, U8, U9, U12, U13, U15, U16	Analog-multiplier AD633
U2, U3, U7, U10, U11, U14, U17, U18, U19	OA TL084

**Table 3 entropy-25-00688-t003:** Variance and standard deviation in histograms.

	Plain	Encrypted	Image in	Image in	Image in	Image in
	Image	Image	Slave 2	Slave 3	Slave 4	Slave 5
α with k=10	3713.12	101.04	3711.78	3711.78	3711.78	3.71178
β with k=10	60.93	10.05	60.92	60.92	60.92	60.92
	**Plain**	**Encrypted**	**Image in**	**Image in**	**Image in**	**Image in**
	**Image**	**Image**	**Slave 2**	**Slave 3**	**Slave 4**	**Slave 5**
α with k=5	3713.12	101.04	96.91	96.87	96.91	96.93
β with k=5	60.93	10.05	9.84	9.84	9.84	9.84

**Table 4 entropy-25-00688-t004:** Structural similarity index.

*P*	*E*	SSIM with k=10	SSIM with k=5
Plain image	Plain image	1	1
Plain image	Encrypted image	0.0030	0.0030
Plain image	Image in slave 2	0.9998	0.0135
Plain image	Image in slave 3	0.9998	0.0134
Plain image	Image in slave 4	0.9998	0.0135
Plain image	Image in slave 5	0.9998	0.0134

**Table 5 entropy-25-00688-t005:** Pearson correlation coefficient.

Coupling	Plain	Encrypted	Image in	Image in	Image in	Image in
Constant	Image	Image	Slave 2	Slave 3	Slave 4	Slave 5
k=10	0.8757	0.1255	0.8520	0.8520	0.8520	0.8520
k=5	0.8758	0.1256	0.1060	0.0957	0.1060	0.0957

**Table 6 entropy-25-00688-t006:** Information entropy results.

Coupling	Plain	Encrypted	Image in	Image in	Image in	Image in
Constant	Image	Image	Slave 2	Slave 3	Slave 4	Slave 5
k=10	7.5250	7.9903	7.5253	7.5253	7.5253	7.5253
k=5	7.5250	7.9903	7.9910	7.9910	7.9910	7.9910

**Table 7 entropy-25-00688-t007:** Decryption error test.

*P*	*D*	E (%) with k=10	E (%) with k=5
Plain image	Encrypted image	99.5688	99.5688
Plain image	Image in slave 2	0.0044	99.6488
Plain image	Image in slave 3	0.0044	99.6488
Plain image	Image in slave 4	0.0044	99.6488
Plain image	Image in slave 5	0.0044	99.6488

## Data Availability

The data used to support the findings of this study are included within the article.

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
