# Peer review of "Network Synchronization of MACM Circuits and Its Application to Secure Communications"

_entropy, 2023, doi:10.3390/e25040688_

Round 1

Reviewer 1 Report

To help authors improve the quality of the manuscript, my comments or suggestions are listed as follows.

1. The font size in some figures is very small. The size of the figures should be further adjusted to the appropriate size. As shown in Figure 1 and Figure 10, it is difficult to see the text clearly.

2. The punctuation after the formula should be checked carefully. For example,  the formulas 9 and 10 should have only one comma, and as well as the formula 11 should be a period.

3. The position and format of subtitles in the figure are not uniform. For example, "(a)" is below the sub-figure in Figure 2, and "a)" is above the sub-figure in Figure 5. It is recommended to uniformly place "(a)" at the bottom of the sub-figure.

4. In lines 168 to 177, it is not recommended to use the equal sign continuously. Authors can consider using set symbols for assignment.

5. There are too few sampling points in the correlation diagram of Figures 21 and 22. Sampling points should be appropriately increased to make the correlation clearer.

6. The analysis index of encryption algorithm is too few. It is expected to add evaluation indicators to comprehensively analyze the algorithm.

7. There are too many references in the same position, such as lines 14, 16, 19 and 23. And most of references are too outdated. Authors can make literatures again and update them with some state-of-the-art references.

8. The abstract section lacks purpose, and the conclusion section should be a paragraph. Authors can refer to the following website: https://writingcenter.unc.edu/tips-and-tools/. Therefore, the abstract and conclusion sections should be polished.

Author Response

Dear Reviewer 1.

By means of this letter, I am sending you the revised manuscript with Reference: Entropy-2255577 (Research Article), which has been submitted for possible publication in Synchronization in Time-Evolving Complex Networks. In this new version of the manuscript, we have incorporated the suggestions and valuable comments provided by the anonymous reviewers.

I thank you in advance for all the attentions you may give to this letter.

My best regards and wishes,

Miguel Ángel Murillo-Escobar
Corresponding author:
CICESE, Department of Electronics and Telecommunications, and  
Engineering, Architecture and Design Faculty, Autonomous University of Baja California (UABC)
Tel: 52 (646)152-82-44. Fax: 646-174 4333
E-mail: [email protected]

Reviewer 2 Report

Dear Authors,

In my opinion, your work has much merit, I highlight your theoretical simulations and applications. However, I think there could be more theoretical sustainability in your results.

See the attached document.

Best regards.

Author Response

Dear Reviewer 2.

By means of this letter, I am sending you the revised manuscript with Reference: Entropy-2255577 (Research Article), which has been submitted for possible publication in Synchronization in Time-Evolving Complex Networks. In this new version of the manuscript, we have incorporated the suggestions and valuable comments provided by the anonymous reviewers.

I thank you in advance for all the attentions you may give to this letter.

My best regards and wishes,

Miguel Ángel Murillo-Escobar
Corresponding author:
CICESE, Department of Electronics and Telecommunications, and  
Engineering, Architecture and Design Faculty, Autonomous University of Baja California (UABC)
Tel: 52 (646)152-82-44. Fax: 646-174 4333
E-mail: [email protected]

Round 2

Reviewer 1 Report

I am satisfied with authors' revisions. I have not further suggestions. It is can be accepted at present.

Reviewer 2 Report

Dear authors,

Your work has merit to be published in Entropy Journal. Your changes in response to the reviewers' comments and suggestions have significantly increased the quality of your work.